# Hemolytic Activity, Cytotoxicity, and Antimicrobial Effects of Silver Nanoparticles Conjugated with Lincomycin or Cefazolin

**DOI:** 10.3390/ijms232213709

**Published:** 2022-11-08

**Authors:** Dmitriy Korolev, Michael Shumilo, Galina Shulmeyster, Alexander Krutikov, Alexey Golovkin, Alexander Mishanin, Anna Spiridonova, Olga Kulagina, Michael Galagudza

**Affiliations:** 1Institute of Experimental Medicine, Almazov National Research Centre, 197341 Saint-Petersburg, Russia; 2Laboratory of Biophysics of Blood Circulation, Pavlov First Saint-Petersburg State Medical University, 197022 Saint-Petersburg, Russia; 3Department of Micro-and Nanoelectronics, Saint-Petersburg Electrotechnical University “LETI”, 197376 Saint-Petersburg, Russia; 4Laboratory of Infiltrative Heart Disease, Almazov National Research Centre, 197341 Saint-Petersburg, Russia; 5Laboratory of Cardiomyopathy, Pavlov First Saint-Petersburg State Medical University, 197022 Saint-Petersburg, Russia; 6Institution of Molecular Biology and Genetics, Almazov National Research Centre, 197341 Saint-Petersburg, Russia; 7Department of Clinical Microbiology, Pavlov First Saint-Petersburg State Medical University, 197022 Saint-Petersburg, Russia; 8Federal State Institution Saint-Petersburg Pasteur Research Institute of Epidemiology and Microbiology, 197101 Saint-Petersburg, Russia; 9Department of Pathophysiology with Clinical Pathophysiology Course, Pavlov First Saint-Petersburg State Medical University, 197022 Saint-Petersburg, Russia

**Keywords:** silver nanoparticles, chemical synthesis, hemolysis, cytotoxicity, antimicrobial activity, antibiotic, conjugates

## Abstract

The overuse of antibiotics has led to the emergence of resistant bacteria. A good alternative is silver nanoparticles, which have antibacterial activity against Gram-negative and Gram-positive bacteria, including multidrug-resistant strains. Their combination with already known antibiotics has a synergistic effect. In this work, we studied the synthesis of conjugates of silver nanoparticles with two antibiotics, lincomycin and cefazolin. Albumin and glutathione were used as spacer shells with functional groups. The physicochemical properties of the obtained conjugates, their cytotoxicity and synergism of antimicrobial activity were studied. The 50% antimicrobial activity of the obtained samples was shown, which allows them to be recommended for use as topical drug preparations.

## 1. Introduction

Silver nanoparticles (AgNP) are a good antimicrobial agent capable of fighting bacteria that cause infections in vitro and in vivo. The antibacterial ability of AgNP extends to Gram-negative and Gram-positive bacteria, including multidrug-resistant strains [1,2].

To improve the biocompatibility of AgNP, they are coated with various shells [3,4,5]. There is also an approach in which, to improve biocompatibility, AgNP are incorporated into a matrix of inorganic substances, such as mesoporous silica [6]. An interesting solution is the shell, where the erythrocyte membrane is used as such [7]. In addition to improving biocompatibility, it allows the introduction of various substances, such as folic acid, which is a targeted agent for cancer cells. These nanoobjects have good biocompatibility. They are able to evade the clearance of the reticuloendothelial system. Peptide ligands are often used as shell materials, which transform nanoparticles into biocompatible materials [5]. Various peptides can simultaneously serve as spacers for the immobilization of other active substances (for example, glutathione) and as targeting ligands for targeted delivery (for example, the γ-ECG tripeptide). Such substances are able to self-assemble into aggregates on the surface of ligand-free silver nanoparticles through intermolecular hydrogen bonds and form shells several nanometers thick. It is possible to obtain several layers of glutathione in close packing based on dimers of carboxylic acids with glycine hydrogen bonds and intermolecular salt bridges between zwitterionic γ-glutamyl groups [5]. Aggregation-induced emission luminogens (AIEgens) are used to synthesize shells of silver nanoparticles. Such core-shell nanoparticles can act as theranostic nanoparticles, allow luminescence, photoacoustic imaging, and radiation therapy, causing cancer cell death [8]. To incorporate doxorubicin into the AgNP shell matrix, organic polymers such as poly(aspartic acid) and poly(ε-caprolactone) are used [9]. Such nanoobjects show increased activity against cancer cells.

Sometimes organic shells enhance the antibacterial effect of AgNP, such as a shell made of chitosan [3] or albumin [4]. The application of inorganic shells of biologically active substances, such as copper oxide Cu_2_O [10], also enhances the antimicrobial effect.

Over the past few decades, the overuse of antibiotics has led to the emergence of resistant bacteria and environmental problems. Both silver nanoparticles and antimicrobial peptides (AMP) could potentially replace antibiotics. Combining AMP and AgNP into a composite material can create new properties such as increased antibacterial activity, lower cytotoxicity, and good stability in an aqueous drug [11,12]. Thus, in [12], a peptide of 13 amino acids with two functional regions was developed: one for antibacterial activity, and the other for the reduction and stabilization of AgNP containing cysteine residues at its C-terminus. As a result, AgNP nanoparticles protected by an antibacterial peptide were synthesized.

One of the most promising directions in the development of antimicrobial drugs is the combination of AgNP with antibiotics, which has a synergistic effect. The effect was substantiated in [13], where a systematic study was carried out to quantify the synergistic effect of antibiotics with different mechanisms of action and different chemical structures in combination with AgNP against *Escherichia coli*, *Pseudomonas aeruginosa*, and *Staphylococcus aureus*. In all, 16 Inhibition of Cell Wall Synthesis antibiotics, 6 Inhibition of Protein Synthesis antibiotics, 4 Inhibition of Nucleic Acid Synthesis antibiotics, and 1 Alteration in Cytoplasmic Membrane—Colistin antibiotic were tested. When using the microdilution method, strong synergistic effects were shown for all tested antibiotics in combination with AgNP at very low concentrations of both antibiotics and AgNP. The effect was independent of the mechanism of action and structure of the antibiotic in combination with AgNP, indicating a non-specific synergistic effect. It has been noted that a very small amount of silver is required for the effective antibacterial action of antibiotics, which is an important finding for potential medical applications due to the negligible cytotoxic effect of AgNP on human cells at these concentration levels. The selective properties of a mixture of AgNP with various antibiotics were also found [14]. The highest synergistic effect, determined by the fractional inhibition index, was shown by the AgNP conjugate with the antibiotics kanamycin and tetracycline against *Staphylococcus aureus*, while AgNP-ketoconazole showed the highest sensitivity in the case of *M. furfur.* Ref [15] studied the selective synergistic antibacterial activity of combined silver nanoparticles with tetracycline, neomycin, and penicillin against the multidrug-resistant bacterium *Salmonella typhimurium* DT104. Dose-dependent growth inhibition of *Salmonella typhimurium* DT104 observed for the combination of tetracycline-AgNP and neomycin-AgNP. The combination of penicillin-AgNP did not cause inhibition.

Previously, there have been attempts to obtain synergistic antimicrobial effect between silver nanoparticles and antibiotics using their mixtures. To the best of our knowledge, chemical conjugation of functionalized AgNPs with commonly used antibiotics, such as cefazolin (CEZ) and lincomycin (LCM), has not been described before. Therefore, the aim of the present work was the synthesis of silver nanoparticles and their conjugation with CEZ and LCM, the study of the physicochemical and biological properties of the obtained conjugates to test the effect of synergistic enhancement of the antimicrobial action. Among the antibiotics frequently used in clinical practice, we chose those that have functional groups that allow their immobilization on the spacer of nanoparticles. Glutathione (GSH) and albumin (Alb), respectively, were used as spacers for CEZ and LCM immobilization.

## 2. Results and Discussion

### 2.1. Immobilization of Antibiotics on AgNPs

Glutathione has traditionally been used as a shell or spacer with a functional group for silver and gold nanoparticles in the development of delivery systems and sensor systems [5,16,17,18].

GSH contains a thiol group that is capable of bonding with silver. Next to the thiol group in the main chain is a secondary amine, which is able to form a hydrogen bond with a silver nanoparticle [19]. Apparently, both functional groups are involved in the reaction with the AgNP surface (Figure 1a,b). The first carboxyl group of GSH is most likely protonated by the primary amine next to it, so it is not active. In this case, a spacer is formed, which also contains two functional groups—a secondary amine and a carboxylate. Cefazolin also has a secondary amine and a carboxyl group that does not react with the AgNP surface. Therefore, the reaction of the glutathione spacer with CEZ most likely proceeds via two competing mechanisms (Figure 1a,b).

Albumin has a stabilizing effect for silver nanoparticles in colloidal solution, forming a protein crown. At the same time, being a transport protein, it contains a large number of different functional groups in the side chains of its constituent amino acids. Usually these two facts are used for immobilization on the surface of silver nanoparticles of various chemical substances, or are used as sensors [20,21].

LCM can be immobilized on the Alb–AgNP framework obtained as follows. In the cysteine (Cys)—silver ion system, extended molecular structures such as a spatial network may appear. The building material for the formation of the network are molecular chains formed as a result of the interaction of cysteine and silver ions (Figure 2a). Thus, a significant increase in the size of globules according to DLS data during the interaction of albumin with silver nanoparticles can be explained. This fact can also be used to construct a scaffold on which the LCM is subsequently immobilized (Figure 2b). Lincomycin was immobilized on monocarboxylic diamino acids included in the albumin chain, such as lysine (Lys) or arginine (Arg).

### 2.2. Physicochemical Properties of Unmodified and Antibiotic-Bound AgNPs

The study of the absorption spectra of the silver nanoparticles colloidal solution (Figure 3) showed that the absorption bands corresponding to the plasmon effect (maximum 434 nm) correlated with those previously described in the literature [22,23].

A transmission electron microscopy (TEM) study of the morphology of the obtained nanoobjects showed that the colloidal solution contains spherical silver particles about 50 nm in size (Figure 4a).

Figure 4b showed that the surface of nanoparticles had thin shell, which was created by glutathione and cephazolin. The average size of these particles was 50 nm.

On Figure 4c the shell had a distinctly wider and fuzzier character, which may be due to the large molecular weight of the albumin protein. The size of these particles was about 50 nm.

The average AgNP size according to the DLS results is 26.37 nm (Figure 5a) with a polydispersity index of 0.5452. There may be a mismatch between the average particle size determined by the DLS and TEM methods due to the over-reflection of the rays in the DLS method, because this method is indirect. When AgNP is modified with a glutathione spacer, the average size of nanoparticles increases to 112.3 nm with a polydispersity index of 0.2003. This increase in average size and decrease in PDI is an indication of the formation of several bound GSH layers on the AgNP surface or the possible crosslinking of several particles. Immobilization of CEZ on the GSH spacer further increases the particle size to 444.1 nm and broadens the distribution with a polydispersity index of up to 0.4785. This increase also indicates the formation of complex chains and the crosslinking of individual particles in solution.

The zeta potential of AgNP according to the ELS results is −46.56 mV (Figure 5b) and decreases in modulus with GSH modification to −24.56 mV, but the colloidal solution still remains stable. These zeta potential values indicate good colloidal stability of the suspensions.

Modification of AgNP with albumin results in an increase in the nanoparticle size to 41.06 nm (Figure 5c) with a polydispersity index of 0.3354. Such a slight increase in size speaks in favor of the formation of a cross-linked protein shell on the AgNP surface and the absence of cross-linking of several particles in solution. Immobilization of lincomycin gives an average particle size of 260.7 nm with a polydispersity index of 0.2633, which indicates the crosslinking of several particles into an agglomerate.

According to the ELS data, the electrokinetic potential of AgNPs modified with Alb decreases in absolute value to −18.41 mV and, upon immobilization of LCM, to −4.96 (Figure 5d), which indicates a decrease in the number of charged functional groups. Despite this, the suspensions retain colloidal stability.

### 2.3. Biological Properties

#### 2.3.1. Hemolytic Activity

The study of the hemolytic activity of AgNPs samples (Table 1) did not reveal a negative effect of AgNP, AgNP–GSH, AgNP–Alb, AgNP–GSH–CEZ on whole blood obtained from healthy donors. It was found that the coefficient of hemolysis of these samples was indistinguishable from the background level and did not exceed 1%. A large hemolytic activity was registered in the AgNP-Alb-LCM sample. The value of the coefficient of hemolysis of this sample after 1 h exceeded 10%, after 24 h—20%. Moreover, the values of the coefficient of hemolysis of this sample for the 1st donor with B III blood group are several times higher. Apparently, this is due to the properties of the antibiotic LCM. Also, when AuNPs are added to LCM, the hemolysis rate decreases, but in the case of CEZ it remains almost unchanged.

#### 2.3.2. Cytotoxicity

##### Results of Fluorescence Microscopy—Quantitative Analysis

A significant decrease in the number of cells on the glass surface was determined compared to the corresponding control group when AgNP–GSH–CEZ and AgNP–Alb–LCM preparations were added to the culture medium at a concentration of 1% and 10%. The decrease was more pronounced in the case of 10% solutions (*p* < 0.05). Whereas, there were no significant differences (see Table 2) in the number of adhered cells when PBS in the same concentration was added (*p* > 0.05).

##### Results of Fluorescence Microscopy—Qualitative Analysis

The cells on the coverslips of the control groups were spread out on the surface of the glasses with the formation of a confluent/subconfluent monolayer (Figure 6), had a typical elongated shape with multiple processes, some of the cells were in the process of division demonstrating proliferation. Longitudinal linear structures, stained red, are clearly visible in the cells—actin microfilaments.

Cells in the groups with the addition of 1% solutions of AgNP–GSH–CEZ and AgNP–Alb–LCM preparations were spread on the glass surface with the formation of a monolayer with a confluence of up to 50–75%, had a typical elongated shape with multiple processes, well-defined longitudinal striation, some cells were in the process of division.

Single irregular/rounded cells with diffusely stained cytoplasm were visualized on the glass surface in groups with the addition of 10% solutions of AgNP–GSH–CEZ and AgNP–Alb–LCM preparations. The cell area was significantly smaller compared to the cells of the control groups and groups with the addition of 1% drug solutions.

Thus, it can be stated that both studied drugs have dose-dependent cytotoxicity, which is slightly more pronounced in AgNP–Alb–LCM.

#### 2.3.3. Antimicrobial Activity

The results of the antimicrobial activity of clear silver nanoparticles were confirmed in previous experiments [4].

The results of the antimicrobial activity of AgNPs with antibiotics are shown in Figure 7 and Figure 8 and the growth inhibition zone (GIZ) are shown in Table 3.

In previous experiments zones of growth inhibition of unmodified AgNPs were about 10–18 mm against different strains [4]. In the presence of antibiotics, zones of growth inhibition increased 2–3 times.

As can be seen from the data presented in the Table 3 and Figure 7 and Figure 8, sample # 1 and # 2 had antimicrobial activity against the cultures tested, this was confirmed by the stunting zones (SZs) of the microorganisms. The GIZ with undiluted antibiotic are approximately equal to the GIZ of silver nanoparticles with antibiotics (coincidence of diameters of the stunting zones of *S. aureus* ATCC 29213, *E. coli* ATCC 25922 and *K. pneumonia* ATCC 13883 around holes with test samples and antibiotics). The antibiotic concentration of silver nanoparticles is half the concentration (concentration of CEZ—50 mg/mL and concentration of LCM—150 mg/mL) of undiluted antibiotic (concentration of CEZ—100 mg/mL and concentration of LCM—300 mg/mL), from which we concluded that the synergistic effect of using silver nanoparticles with antibiotic enhances the antibiotic by about two times. In clinical practice this can reduce the prescribed dose of antibiotics.

The data obtained in the present study have demonstrated the presence of robust antibacterial effects of AgNPs conjugated with two commonly used antibiotics. It should be noted that native AgNPs have been extensively studied before as antimicrobial agents [24]. Antibacterial effects of prototypical AgNPs are usually attributed to silver ion release from well-developed nanoparticle surface, resulting in inactivation of crucial bacterial enzymes and proteins due to formation of silver thiolate complexes [25]. Other putative mechanisms of Ag^+^ antibacterial activity include interaction with nucleosides within the bacterial DNA and formation of cytotoxic amounts of reactive oxygen species. In addition to antibacterial activity, AgNPs have been recently shown to possess potent antiviral effects, although the mechanisms of AgNPs-mediated virus inhibition are poorly studied [26]. It is well known that the extent of biocidal effect of AgNPs is influenced by such factors as their shape, size, coating type, and the duration of contact between the nanoparticles and bacterial cells. However, bare AgNPs typically demonstrate quite low stability in aqueous solutions, the fact that resulted in the development of multiple types of polymeric AgNP coatings [27]. In particular, albumin coating of 40-nm spherical AgNPs resulted in significant improvement in their stability, reduction of cytotoxicity and presence of potent antimicrobial action [4]. The coating of AgNPs with chitosan and bovine serum albumin has been associated with significant increase in their stability and, in addition, with the presence of marked bactericidal effects against 7 oral and non-oral bacteria, provided that the magnitude of antimicrobial effect has been higher in chitosan-covered AgNPs having smaller diameters [3].

The antimicrobial properties of AgNPs might be further increased by the addition of antimicrobial agents to the coating material. This concept has been already validated for vancomycin [28] and tetracycline [15]. For example, vancomycin-modified AgNPs showed significant antibacterial activity at a low dose against gram-positive (*Staphylococcus aureus*) and gram-negative (*Escherichia coli*) bacteria. In another study, Deng et al. [29] studied the mechanism of action of a mixture of AgNPs with four classes of antibiotics, β-lactams (ampicillin and penicillin), quinolones (enoxacin), aminoglycosides (kanamycin and neomycin), and polypeptides (tetracycline) against multiresistant *Salmonella typhimurium*. It was found that the synergistic effect of the mixture is due to the formation of complex compounds of silver with antibiotics. Our data confirm these earlier observations, demonstrating enhanced antibacterial activity of CEZ and LCM immobilized on AgNPs against four different bacterial species.

## 3. Materials and Methods

### 3.1. Synthesis of Silver Nanoparticles

AgNPs were synthesized by chemical reduction of silver nitrate in an aqueous phase using sodium citrate as reducing agent [30,31]. Briefly, a mixture of 6.25 mL of distilled water, 1.25 mL of sodium citrate (1% by weight), 1.25 mL of silver nitrate (1% by weight) and 50 μL of potassium iodide (300 μM) was prepared by stirring at room temperature and kept for 5 min. This mixture was poured into 237.5 mL of boiling distilled water that included 250 μL of ascorbic acid (0.1 M), and stirred. The color of the solution changed to yellow, then to slightly orange, thus indicating formation of nanoparticles. Then the colloidal solution was boiled for 15 min. After cooling, silver nanoparticles were stored in a dark glass container in a dark place at 4 °C. All reagents were purchased from Sigma Aldrich (Burlington, Massachusetts, USA).

The resulting colloidal solution was purified by dialysis through a 35 kDa membrane against distilled water. The properties of the resulting suspension of AgNPs were described previously in more detail [4].

### 3.2. Engraftment of Spacers and Immobilization of Antibiotics

To modify AgNP, the initial suspension was evaporated twice by volume.

AgNP modification was carried out with 1% GSH solution (Sigma Aldrich, CAS Number 70-18-8). To do this, 5 mL of the resulting suspension of silver nanoparticles was taken in a 15-mL polypropylene tube, 100 μL of glutathione solution was added and stirred for 20 min on an orbital shaker (LS-220, LOIP, Saint Petersburg, Russia) at a stirring speed of 300 min^−1^. The resulting suspension was subjected to dialysis through a membrane with a pore size of 35 kDa.

For immobilization of Alb, 1 mL of Alb solution (Human Albumin 20%, Octapharma, Vienna, Austria) was added to 5 mL of AgNP suspension in a 15 mL polypropylene tube and mixed for 2 h on an LS-220 orbital shaker (LOIP, Saint Petersburg, Russia) at stirring speed 300 min^−1^. The resulting suspension was subjected to dialysis through a membrane with a pore size of 35 kDa.

A solution of cefazolin was prepared by diluting a lyophilizate of a pharmaceutical preparation (LEKKO, Saint Petersburg, Russia). A solution of lincomycin hydrochloride was used ready-made (Dalchimpharm, Khabarovsk, Russia).

To immobilize antibiotics, 5 mL of AgNP-GSH or Ag-Alb were mixed with 5 mL of an antibiotic solution and stirred for 1 h on an orbital shaker at a stirring speed of 300 min^−1^. The resulting suspension was subjected to dialysis through a membrane with a pore size of 35 kDa. All manipulations were carried out at a temperature of 20 °C and atmospheric pressure.

### 3.3. Study of Physical and Chemical Properties of Nanoparticles

The average hydrodynamic diameter of nanoparticles and the polydispersity index (PDI) were studied by dynamic light scattering (DLS), the zeta potential distribution was determined by electrophoretic light scattering (ELS) (Zetasizer Ultra, Malvern Instruments Ltd., Worcestershire, UK).

Transmission electron microscopy (TEM) images were obtained for dry samples (JEM-2010, JEOL, Tokyo, Japan).

Absorption spectra of colloidal solutions were obtained using a spectrophotometer (Unico 2802s, Unico Sys, Dayton, Ohio, USA).

### 3.4. Study of Hemolytic Activity of Colloidal Solutions

The hemolytic activity of the obtained solutions was studied on whole blood of two donors.

Blood of the donor 1: B III Rh + kell −.

Blood of the donor 2: AB IV Rh + kell −.

To increase the osmolarity in the suspension was added 0.9% of the mass of sodium chloride.

0.5 mL of each of the 5 test solutions were incubated with 0.5 mL of blood from each of the donors for 1, 2 and 24 h in a biological thermostat at 37 °C.

Blank samples were also prepared for positive control by adding distilled water (PC), and for negative control by adding saline (NC) hemolysis.

Also were prepared samples with antibiotics.

After incubation, the samples were centrifuged for 20 min at a speed of 2000 min^−1^.

Hemolytic activity was assessed by hemolysis coefficient. The latter was determined spectrophotometrically basing on optical density of the samples at wavelength 415 nm, which corresponds to the absorption band of oxyhemoglobin. The used spectrophotometer was Unico 2802 (Unico Sys, USA).

To measure optical density, 200 μL of the sample was brought up to 6 mL with saline solution.

Hemolytic activity was calculated using the following formula:HC = (O − NC)/(PC − NC) × 100%
where O—is the measured optical density of the sample;

NC—is negative control (zero hemolysis of the blank sample);

PC—is positive control (one hundred percent hemolysis of the blank sample).

### 3.5. Study of Cytotoxic Properties

The experiment was performed using human multipotent mesenchymal stem cells (MMSCs) obtained from the subcutaneous adipose tissue of healthy donors. MMSCs were cultivated in alpha-MEM culture medium (Thermo Fisher Scientific, Waltham, MA, USA) supplemented with 10% fetal bovine serum (FBS), 1% L-glutamine, and 1% penicillin/streptomycin solution (Thermo Fisher Scientific, Waltham, MA, USA) in a CO_2_ incubator at 37 °C and 5% CO_2_. The study was performed according to the Helsinki declaration and approval was obtained from the local Ethics Committee of the Almazov National medical research centre (№ 12.26/2014; 1 December 2014). Written informed consent was obtained from all subjects prior to fat tissue biopsy.

The cytotoxic properties of AgNP-GSH-CEZ and AgNP-Alb-LCM preparations were studied at concentrations of 1% and 10%. The same volumes of phosphate-buffered saline (PBS) added to cell culture medium were used as controls. The experiment was carried out on cover slips 12 mm in diameter in triplicates.

A total of 18 sterile coverslips were placed in the wells of a 24-well plate (4 groups of preparations + 2 control groups with PBS). Then, 1 mL of a suspension of MMSCs at a concentration of 50,000 cells/mL was added to each well and cultured for 24 h. Then, the culture medium was replaced and the studied preparations/PBS were added to the wells of the plate in a volume of 10 μL (1%) and 100 μL (10%), the plate was placed in a CO_2_ incubator and cultivated at 37 °C, 5% CO_2_ and 95% relative humidity. The preparations and PBS were preheated in a water bath, the preparations were additionally vortexed. The cells were cultivated for 48 h, after which the slides were washed three times using PBS and fixed in a 4% paraformaldehyde (PFA) solution for 10 min.

After fixation, the slides were washed three times using PBS and subjected to immunocytochemical staining with rhodamine-labeled phalloidin according to a previously described protocol [4,16,31]. According to the protocol, cover slips with seeded cells were first treated with a 0.05% solution of Triton X-100 for 3 min, followed by washing three times in PBS. Next, a solution of rhodamine-labeled phalloidin (Thermo Fisher Scientific, Waltham, MA, USA) at a dilution of 1:500 in 1% FBS-PBS was added to the wells, incubated for 20 min at room temperature, and then washed five times in PBS. At the final stage, cell nuclei were stained with DAPI (4,6-diamidino-2-phenylindole). To do this, the stock solution of DAPI was diluted in PBS at a ratio of 1:40,000, the working solution was added to the wells and incubated for 40 s, after which the cover slips were thoroughly washed to remove dye residues in PBS.

After staining, the slides were mounted on glass slides using a preparation medium and stored in the dark at 4 °C.

Stained MMSCs on cover slips were subjected to fluorescent microscopy with quantitative and qualitative analysis of adherent cells. The cells were visualized using an Axiovert inverted fluorescent microscope (Zeiss Microscopy, Jena, Germany) and a compatible Canon camera. DAPI fluorescence was recorded using an appropriate filter, rhodamine phalloidin fluorescence was recorded using the Rhodamine channel. Quantitative analysis of cells was performed by counting stained cell nuclei at ×100 magnification in 10 fields of view for each technical replicate, followed by recalculation of the number of MMSCs per 1 mm^2^. Statistical processing of the obtained data was carried out using GraphPad Prism 8 software (version 8.0.2, San Diego, CA, USA) with the nonparametric Mann-Whitney U test. The results were presented as mean (Mean) and standard deviation (SD). When conducting a qualitative analysis, the morphology of cells was assessed by the stained cytoskeleton by analyzing photographs at ×100 and ×400 magnifications.

### 3.6. Assessment of Antimicrobial Activity

The microbiological research was carried out to evaluate the antimicrobial activity of silver preparations.

The following preparations were investigated in the work:

Sample # 1—AgNP–Alb–LCN (sterile solution);

Sample # 2—AgNP–GSH–CEZ (sterile solution);

The antimicrobial action of the studied preparations was assessed using a disk-diffusion agar test [17].

The antimicrobial action of the preparations was studied against the collection strains: *S. aureus* ATCC 29213; *E. coli* ATCC 25922; *K. pneumonia* ATCC 13883; *P. aeruginosa* ATCC 27853.

The inoculum was prepared by direct suspension of daily culture colonies in sterile isotonic solution to a density of 0.5 McFarland turbidity standard, which approximately corresponds to a load of 1.5 × 10^8^ CFU/mL. The optical density of the microbial suspension was monitored densitometrically (densitometer model Densimat “Bio Merieux”, Marcy-l’Étoile, France).

A suspension of test cultures (1.5 × 10^8^ CFU/mL) was applied to the surface of Mueller Hinton agar (BioMedia, Saint Petersburg, Russia) in Petri dishes, then holes of 9 mm diameter were made in the agar. Then, 100 µL of silver preparation was added to test holes (three holes per dish), 100 µL of sterile water for injection from a pharmacy was added to control holes (C) (one hole per dish). The plates were incubated in an incubator at 36 ± 1 °C for 18–20 h, then the results were recorded. The stunting zone around the hole was measured (taking into account the diameter of the hole itself of 9 mm). The experiment was performed in three replicates and the mean value was taken into account.

### 3.7. Statistical Analysis

Statistical processing of the obtained data on AgNP cytotoxicity was performed in the GraphPad Prism 9 software (version 9.3.1) using the Mann–Whitney nonparametric U-test. Results were presented as mean ± standard deviation (SD).

## 4. Conclusions

In conclusion, 50-nm AgNPs were synthesized by chemical reduction, followed by functionalization with either glutathione (GSH, AgNPs-GSH) or albumin (Alb, AgNPs-Alb). Cefazolin (CEZ) or lincomycin (LCM) have been conjugated with functional groups of GSH (AgNPs-GSH-CEZ) or Alb (AgNPs-Alb-LCM), respectively. AgNPs, AgNP-GSH, AgNP-Alb, AgNP-Alb-LCM demonstrated minimal hemolytic activity, while AgNPs-GSH-CEZ produced significant hemolysis after 1 and 24 h of incubation. In addition, both AgNPs-GSH-CEZ and AgNP-Alb-LCM showed dose-dependent cytotoxic effects on human adipose-tissue-derived mesenchymal stem cells with slightly more pronounced cytotoxicity in the latter. The antimicrobial effects of AgNPs-GSH-CEZ and AgNP-Alb-LCM against *S. aureus*, *E. coli*, *K. pneumoniae*, and *P. aeruginosa* were significantly greater than the effects of free antibiotics. We conclude that conjugation of AgNPs with either CEZ or LCM results in improved antimicrobial activity, which may contribute to reduction in the doses of antibiotics and limitation of their side effects.

## Figures and Tables

**Figure 1 ijms-23-13709-f001:**
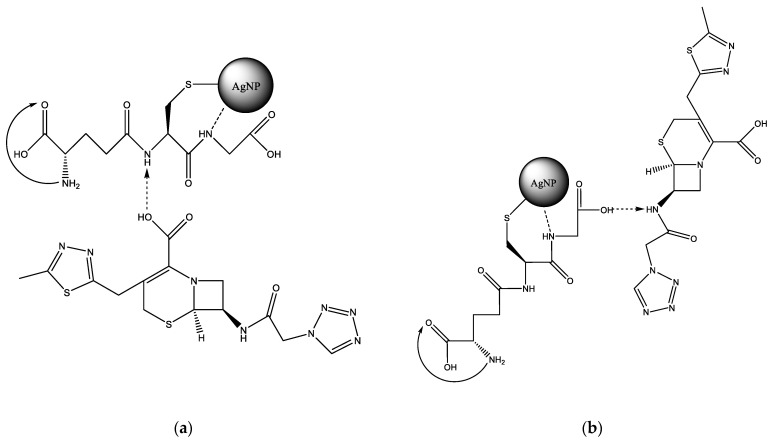
Variations in the reaction of GSH with CEZ with protonation of the secondary amino group: (**a**)—GSH, (**b**)—CEZ.

**Figure 2 ijms-23-13709-f002:**
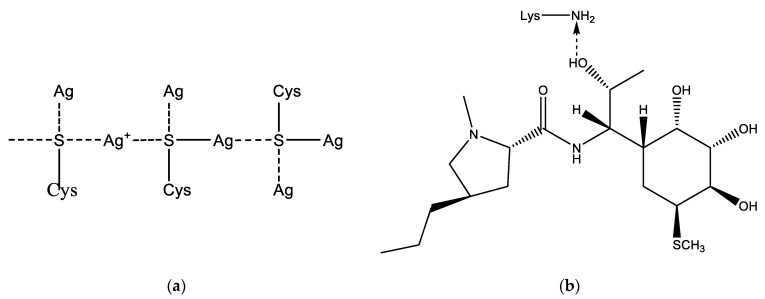
Possible addition reactions of LCM to AgNP: (**a**)—mechanism of cross-linking of albumin with a silver nanoparticle due to sulfur-containing amino acids; (**b**)—interaction of LCM with albumin amino acid lysine.

**Figure 3 ijms-23-13709-f003:**
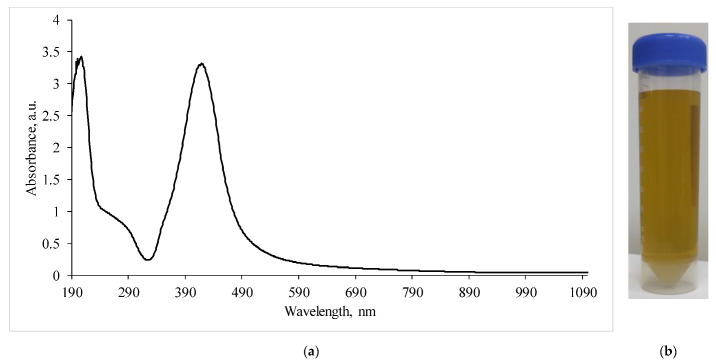
(**a**)—Absorption spectra of a silver nanoparticles suspension diluted by half; (**b**)—visualization of the AgNP suspension.

**Figure 4 ijms-23-13709-f004:**
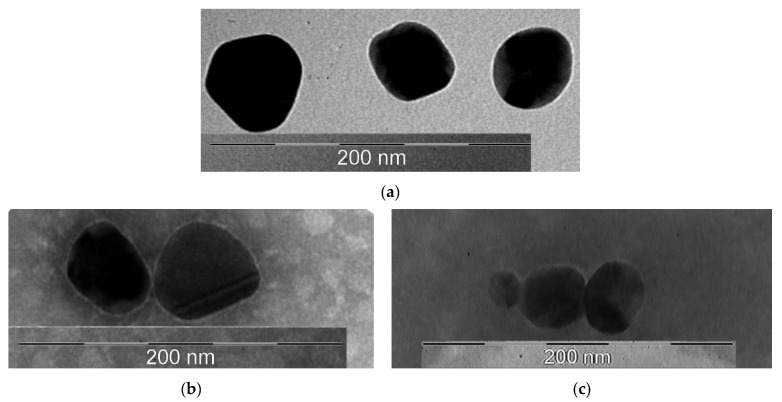
TEM photographs of the obtained samples: (**a**)—AgNP; (**b**)—AgNP–GSH–CEZ; (**c**)—AgNP–Alb–LCM.

**Figure 5 ijms-23-13709-f005:**
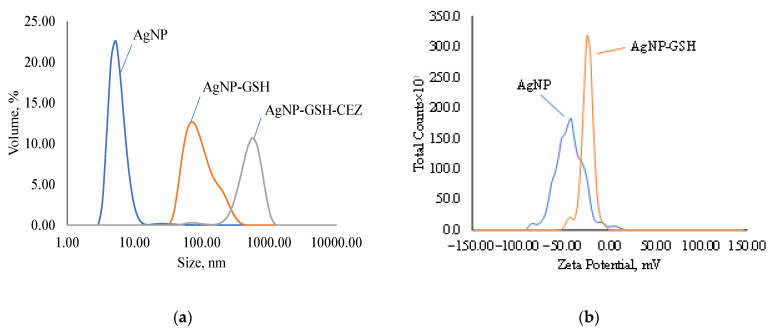
Disperse composition and zeta potential of samples: (**a**)—disperse composition of AgNPs, AgNps with GSH and AgNPs with GSH and CEZ; (**b**)—zeta potential of AgNPs and AgNps with GSH; (**c**)—disperse composition of AgNPs, AgNps with Alb and AgNPs with Alb and LCM; (**d**)—zeta potential of AgNPs and AgNps with Alb.

**Figure 6 ijms-23-13709-f006:**
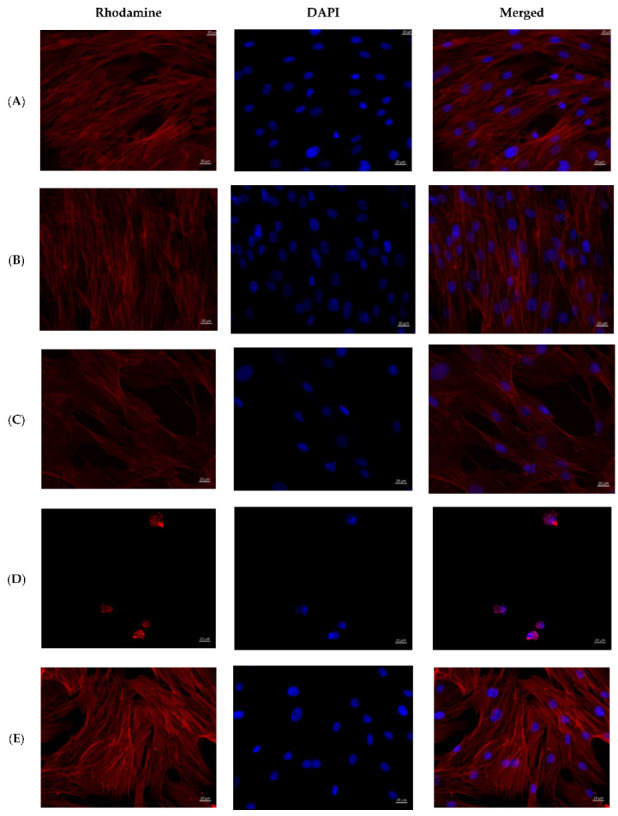
MMSCs after co-cultivation with colloidal silver preparations. Cytoskeleton staining with rhodamine-labeled phalloidin, nuclei staining with DAPI, ×400 magnification. (**A**)—PBS 10 µL; (**B**)—PBS 100 µL; (**C**)—preparation AgNP–GSH-CEZ—10 µL; (**D**)—preparation AgNP-–GSH–CEZ—100 µL; (**E**)—AgNP–Alb–LCM preparation—10 µL; (**F**)—AgNP–Alb–LCM preparation—100 µL.

**Figure 7 ijms-23-13709-f007:**
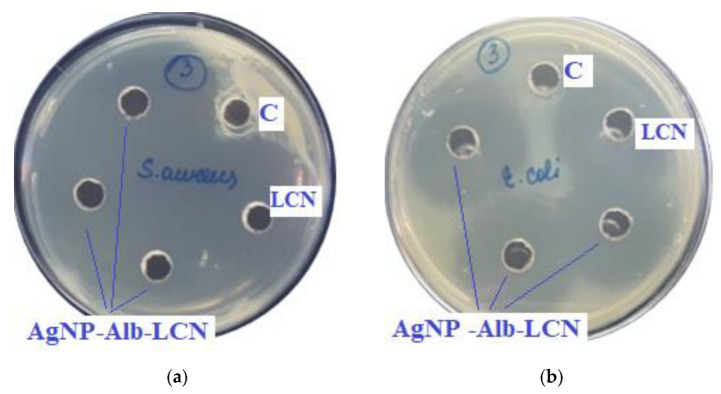
Assessment of the antimicrobial action of AgNP–Alb–LCN (concentration of LCN—150 mg/mL), 100 µL of sample in each hole, against test strains: (**a**)*—S. aureus*, (**b**)*—E. coli*, (**c**)*—K. pneumonia*, (**d**)*—P. aeruginosa*. C—control hole with water for injection, LCN—Lincomycin (concentration—300 mg/mL).

**Figure 8 ijms-23-13709-f008:**
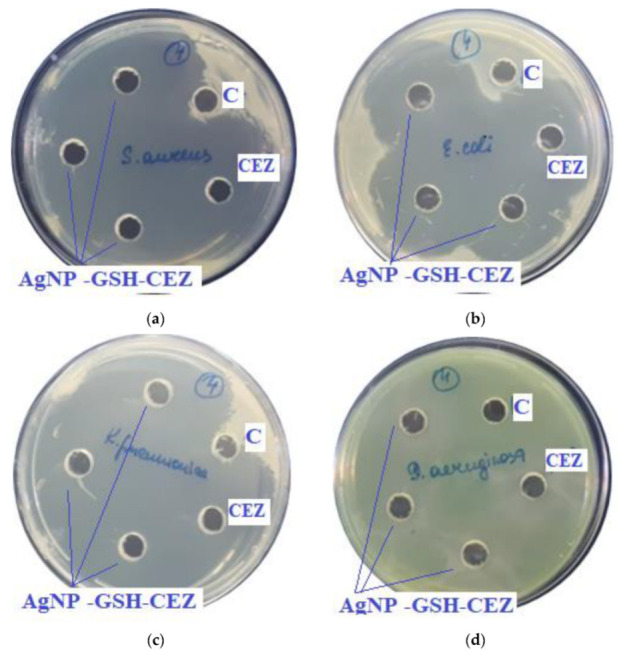
Assessment of the antimicrobial action of AgNP–GSH–CEZ (concentration of CEZ—50 mg/mL), 100 µL of sample in each hole, against test strains: (**a**)*—S. aureus*, (**b**)*—E. coli*, (**c**)*—K. pneumonia*, (**d**)*—P. aeruginosa*. C—control hole with water for injection, CEZ—Cefazolin (concentration—100 mg/mL).

**Table 1 ijms-23-13709-t001:** The hemolysis coefficient measurement results.

Sample	Hemolysis Coefficient, %, Mean ± SD					
	Blood of the Donor 1			Blood of the Donor 2		
	1 h	2 h	24 h	1 h	2 h	24 h
NC	0	0	0	0	0	0
PC	100	100	100	100	100	100
AgNP	0.221 ± 0.009	0.496 ± 0.019	0.589 ± 0.024	0.072 ± 0.003	0	0.068 ± 0.003
CEZ	0.001 ± 0.021	0.733 ± 0.713	0	0	0.072 ± 0.064	0.388 ± 0.004
LCM	86.879 ± 4.748	90.613 ± 1.413	86.718 ± 1.051	54.951 ± 1.487	69.374 ± 3.488	74.684 ± 1.052
AgNP-GSH	0.129 ± 0.005	0.332 ± 0.013	0.898 ± 0.036	0.288 ± 0.012	0.112 ± 0.004	0.138 ± 0.006
AgNP–Alb	0.611 ± 0.024	1.012 ± 0.038	0.457 ± 0.018	0.812 ± 0.032	0.963 ± 0.038	0.557 ± 0.022
AgNP–GSH–CEZ	0.262 ± 0.011	0.222 ± 0.009	0.058 ± 0.002	0.202 ± 0.008	0	0.341 ± 0.014
AgNP–Alb–LCM	80.882 ± 3.238	77.144 ± 3.081	68.442 ± 2.742	12.837 ± 0.506	26.511 ± 1.058	22.009 ± 0.878

**Table 2 ijms-23-13709-t002:** Number of MMSCs nuclei detected on the surface of coverslips after co-cultivation with AgNP preparations.

Preparations	Cells/mm^2^, Mean ± SD
PBS 10 µL	581 ± 83
PBS 100 µL	592 ± 89
AgNP–GSH–CEZ—10 µL	482 ± 80 *
AgNP–GSH–CEZ—100 µL	2 ± 3 *
AgNP–Alb–LCM—10 µL	433 ± 160 *
AgNP–Alb–LCM—100 µL	2 ± 3 *

Significance of differences compared to the corresponding control group (Mann-Whitney): * *p* < 0.05.

**Table 3 ijms-23-13709-t003:** Zones of strain growth inhibition.

Preparation		TEST CULTURES			
		Zones of Growth Inhibition (mm)			
		*S. aureus*	*E. coli*	*K. pneumoniae*	*P. aeruginosa*
Sample # 1—AgNP–Alb–LCN (sterile solution)	Control	9 *	9	9	9
	LCN	50 ± 1	32 ± 1	35 ± 1	18 ± 1
	AgNP–Alb–LCN	50 ± 1	32 ± 2	35 ± 2	12 ± 1
Sample # 2—AgNP–GSH–CEZ (sterile solution)	Control	9 *	9	9	9
	CEZ	45 ± 1	41 ± 1	50 ± 1	9
	AgNP–GSH–CEZ	45 ± 2	40 ± 2	48 ± 2	9

Note: C—control, 100 µL sterile water for injection from a pharmacy was added to the holes. * Absence of microbial stunting zone corresponds to 9 mm—hole diameter.

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
