# Peer review of "Hemolytic Activity, Cytotoxicity, and Antimicrobial Effects of Silver Nanoparticles Conjugated with Lincomycin or Cefazolin"

_ijms, 2022, doi:10.3390/ijms232213709_

Round 1
Reviewer 1 Report
In their work, Korolev et. al., synthesized AgNPs conjugated with Lincomycin and Cefazolin and investigated their hemolytic activity, cytotoxicity and antimicrobial effects. However, the manuscript in the present form have some discrepancies. The following comments should be addressed before acceptance.
1. Please explain the novelty of this work.
2. FTIR should be conducted to investigate chemical bonding/chemical interaction between antibiotics and AgNPs
3. UV-Vis spectra and image of AgNPS dispersed in solution (orange solution) should be added.
4. What kind of DI water has been used? (Type I, Type II or distilled water?). This is importance because the size and shape of AgNPs also depend on pH and ionic strength of water.
5. For hemolytic activity, any human blood sample test requires bioethics approval. Do the author’s research project have bioethics approval for using human blood? If the authors have the approval, please statement in this method.
6. How many replication do the authors test for this assay? If the authors used this formulation, the results in Table 1 (Hemolysis coefficient %) should be mean and sd.
7. Line 148: Is the unit of centrifuge correct (2000 min-1) ?
8. Line 195: Positive sign is not required (+4).
9. Line 312 and Table 1: the authors must confirm the hemolytic test with antibiotics alone because cefazolin is normally used as intravenous injection dosage form for infection treatment, where plentiful of erythrocyte. Although there are some case reports of cefazolin induce hemolytic anemia, the in vitro test is necessary to explain the results. The authors should explained more details how AgNP-GSH-CEZ caused hemolysis in blood of the 1st donor.
10. Figure 6 and 7: the authors should indicated the name of compound and concentration for each hole.
11. How to measure the inhibition zone when there are some overlapped zone or no margin of inhibition zone (figure 6a, 7a and 7c) ?
In the intersection segment, the diffused compounds from two hole are combined and can lead to synergism or antagonism to the test samples, in particular the antibiotic hole near by the test sample. The statement in lines 368-371 will be clear if there is no overlapping zone.
12. In Table 3, the results of antibacterial test in table 3 is not well designed, please revise it.
13. Line 363: there is no Figure 8.
14. Many of scientific names of species are not written in italic. Please correct them.
15. There are few typos in the manuscript, please check it carefully.
Reviewer 2 Report
This work reported the synthesis of conjugates of silver nanoparticles with two antibiotics, lincomycin, and cefazolin. Albumin and glutathione were used as spacer shells with functional groups. Combining AMP and AgNP into a composite material can create new properties such as increased antibacterial activity, lower cytotoxicity, and good stability in an aqueous drug. In addition, a very small amount of silver is required for the effective antibacterial action of antibiotics, which is an important finding for potential medical applications due to the negligible cytotoxic effect of AgNP on human cells at these concentration levels. Works synthesized silver nanoparticles and their conjugation with known antibiotics-cefazolin (CEZ) and lincomycin (LCM) and studied the mechanism of action of a mixture of silver nanoparticles with four classes of antibiotics, β-lactams (ampicillin and penicillin), quinolones (enoxacin), aminoglycosides (kanamycin and neomycin), and polypeptides (tetracycline) against multiresistant Salmonella typhimurium bacterium. It is quite possible that conjugation of AgNPs with either CEZ or LCM results in improved antimicrobial activity, which may contribute to the reduction in the doses of antibiotics and limitation of their side effects. Here, I have some questions as follows.
1. Please explain the detailed influence of the temperature in the reaction of the direct addition of LCM to silver nanoparticles.
2. Can the crosslinking of several particles into agglomerate be observed through SEM images?
3. How to calculate the coefficient of hemolysis of these samples?
4. The threshold of the dose-dependent cytotoxicity of the AgNP-Alb-LCM.
5. Can the antimicrobial activity of AgNPs with antibiotics be confirmed by other bacterial?
6. Some related works should be cited. CrystEngComm, 2013, 15, 7230–7235. Chemical Engineering Journal 429 (2022) 132342.
Reviewer 3 Report
Following are the comments that need be addressed in the revised version.
1. Abstract should be more concise with some numbers such as " high antibacterial activity" is a very generic statement, please be specific and present values in percentage as well.
2. Did authors performed the antibacterial activity of the two antibiotics, lincomycin and cefazolin separately? I can see only 3 samples, (Sample â„–1 - Silver nanoparticles - AgNP (sterile solution); Sample â„–2- AgNP -Alb-LCN (sterile solution); and Sample â„–3 AgNP -GSH-CEZ (sterile solution);). It is suggested that authors should include the test from only antibiotics utilized n this study, this will give a clear idea about the effect of silver nanoparticles.
3. Whole results and discussion section contains only one reference, although there are so many literature available related to antibacterial activity of sliver nanoparticles as well as antibiotics utilized in this study, results should be compared properly with the appropriate literatures.
4. The size of the silver nanoparticles also plays important role in the efficacy of its antibacterial activity, did authors tried to see effect of AgNPs size on the antimicrobial efficacy? what was the reason behind selecting this size range?
Reviewer 4 Report
In this paper, the author has studied the synthesis of conjugates of silver nanoparticles with two antibiotics, cefazolin (CEZ) and lincomycin (LCM), and further investigated the physicochemical and biological properties of the resultant conjugates to test the effect of synergistic enhancement of the antimicrobial action. Overall the paper is well-structured and organized. However, the authors should consider major corrections before the manuscript can be published. Some comments and suggestions are listed as follows:
1. Author incorporates the AgNPs into the known Antibiotic to get an enhanced effect of antibiotics. But the author did not include the only antibiotic to see if the antimicrobial effect enhanced or not. I will suggest that the author do a few experiments in the presence of only antibiotics and compare the result with the AgNPs-Antibiotic conjugate.
2. Why did the author specifically choose GSH and Albumin as spacers? Need explanation in result and discussion part.
3. Figure 2: I was wondering how the authors confirmed that one CH4 gr is eliminated after adding AgNPs to LCM. Please explain.
4. I will suggest the author to redraw the Figures according to the journal style.
5. Conclusion is not well written. I will suggest the author rewrite the conclusion. The first two paragraphs of the conclusion should go to the introduction part.
Round 2
Reviewer 1 Report
Dear Authors,
Some responses is still not quiet clear. Here are my opinion for the Author’s responses.
Comment no. 2
Point: “FTIR should be conducted to investigate chemical bonding/chemical interaction between antibiotics and AgNPs”
Response: In our opinion, FTIR will not show chemical bonds, but will only show the presence of functional groups. Therefore, we have not performed FTIR in this study. Unfortunately, it is not possible to perform additional measurements now. So, we apologize for the inability to add this information.
Opinion: FTIR experiment can probe the chemical bonding via the shift of vibration peak of each functional group such as SH, NH, OH and COOH. So, FTIR spectra will help to confirm exact chemical bonding in Figure 1 and 2.
Comment no 6.
Point: “How many replication do the authors test for this assay? If the authors used this formulation, the results in Table 1 (Hemolysis coefficient %) should be mean and sd.”
Response: We used three replications, so we added information in Table 1.
Opinion: Three replication is fine. However, the significant figures or number of decimal places should be consistency in the same table. And is it possible to report in bar chart or line chart to make it more readable?
Comment no. 8.
Point: “Line 312 and Table 1: the authors must confirm the hemolytic test with antibiotics alone because cefazolin is normally used as intravenous injection dosage form for infection treatment, where plentiful of erythrocyte. Although there are some case reports of cefazolin induce hemolytic anemia, the in vitro test is necessary to explain the results. The authors should explained more details how AgNP-GSH-CEZ caused hemolysis in blood of the 1st donor.”
Response: In fact, we’ve performed experiments with pure antibiotics (CEZ and LCM). These data have been added to the text (Table 1).
Opinion: In Table 1, the % hemolysis coefficient of CEZ was significantly increased when it was formed AgNP-GSH-CEZ, whereas the hemolysis coefficient of AgNP-Alb-LCM was significantly decreased when it was formed. The author must discussed more and give the possible mechanism for this phenomena and add more reference.
Comment no. 10
Point: “Figure 6 and 7: the authors should indicated the name of compound and concentration for each hole.”
Response: In each hole was added 100 μl of sample. We added this information to Figures 7,8.
Opinion: A 100 uL of sample was addressed in the method, so it is not necessary to mention again. However, the concentration must be added because the author referred to the concentration in line 390-397.
Comment no. 12
Point: “In Table 3, the results of antibacterial test in Table 3 is not well designed, please revise it.”
Response: We redesigned the table.
Opinion: The results in Table 3 is not agreed with the method in page 5 section 2.6 line 219 and 222. Why there is no result of sample #1 AgNP (sterile solution)? The author must include this in Table 3.
Reviewer 2 Report
I am satisfied with the reply letter and I suggest the publication of this work without modification.
Author Response
I, along with my co authors, thank you for consideration of our manuscript
Reviewer 4 Report
The author has addressed all the comments. The manuscript can be accepted in its present form.
Author Response

(The authors gave the same response as above.)

Round 3
Reviewer 1 Report
The second revised version is not suitable to publish in IJMS. I still find out more inaccuracy and make this manuscript seem to be unreliable. For examples, in the 1st version (Line 212), there are 3 investigated samples and they not include sample #1 AgNPs in the result, however in the 2nd version, why AgNPs was deleted? In the 2nd version 3.3.1 Hemolytic activity, line 329 "AgNP-Alb-LCM" was replaced with "AgNP-GSH-CEZ" and in line 336, where is AuNPs come from. Furthermore, Table 2, row 5 and 7 the Mean ± SD was 2±3.? The author should not just response to reviewer’s comments without carefully proof-reading the manuscript. This lead to some question that have all authors read the entire revised manuscript before submission?